# Recurrent Neural Network Architecture based on Dynamic Systems Theory for Data Driven modelling of Complex Physical Systems

## Abstract

While dynamic systems can be modeled as sequence-to-sequence tasks by deep learning using different network architectures like DNN, CNN, RNNs or neural ODEs, the resulting models often provide poor understanding of the underlying system properties. We propose a new recurrent network architecture, the Dynamic Recurrent Network (DYRNN), where the computation function is based on the discrete difference equations of basic linear system transfer functions known from dynamic system identification. This results in a more explainable model, since the learnt weights can provide insight on a system's time dependent behaviour. It also introduces the sequences' sampling rate as an additional model parameter, which can be leveraged, for example, for time series data augmentation and model robustness checks. The network is trained using traditional gradient descent optimization and can be used in combination with other state of the art neural network layers. We show that our new layer type yields results comparable to or better than other recurrent layer types on several system identification tasks.

## 1 Introduction

Dynamic systems occur in many different areas of life (Isermann & Münchhof (2011)). From biology, engineering, medicine to economics and more: Often, if a system changes its state based on a external input, this system can be viewed as a dynamic system. Dynamic system identification is the process of modelling the system's properties. Such models can be used, for example, for anomaly detection, controller design or outcome prediction. For linear systems, this identification task is already well understood and state of the art methods exist.

However, if a system exhibits non-linear behaviour, for example slip-stick-effects due to mechanical friction, the applicability of these methods is limited. In this case different approaches implemented in the state of the art range from white-box to black-box models. Generally, increasing system complexity raises the need for more powerful and often less understandable model architectures in order to produce satisfactory results: White box (based on differential equations or numerical simulations of the physical system components), black box systems (like Gaussian processes, deep neural networks, Support Vector Machines) and grey box models, which often employ a mix of linear and non-linear building blocks.

One example of a tool used in engineering are Hammerstein-Wiener models which are a combination of linear and (prior known) non-linear equations (shown in Figure 1). The linear model parameters are determined based on the training data. The non-linear behaviour of models is modeled using lookup tables or user defined non-linear functions.

In this work we present a new type of recurrent neural network layer called the Dynamic Recurrent Neural Network (DYRNN). It is designed for data based modelling of dynamic systems in a sequence-to-sequence manner based on input $(x(t))$ and output $(y(t))$ data. With it, we intend to bridge the gap between dynamic systems theory and recurrent neural networks. The layer's internal computation is based on elemental transfer blocks from linear system identification. By combining it with non-linear neural networks, a Hammerstein-Wiener style model is emulated. This way, the

model can offer additional knowledge about the examined system's internal properties. Furthermore, while the model is trained on sampled data of one sampling rate it can be applied to data of the same system at a different sampling rate. This can be used to check the robustness of the model or to save time during training. We show that our network produces results which are better than or comparable to other recurrent networks (RNN, LSTM, GRU) on three different problem datasets. Since the layer can be implemented to be compatible to current deep learning frameworks, it can be combined with state of the art neural network layers (like convolutional or fully connected layers) and training techniques.

Figure 1: Hammerstein-Wiener model. Static non-linearities before and after a linear differential equation model $g(t)$ can be used to model non-linear dynamic systems.

## 2 RELATED WORK

Dynamic system identification can be viewed as a sequence-to-sequence task of the modelling of a systems' output based on certain inputs. Isermann & Münchhof (2011), for example, list several different tools like ARIMA processes for linear systems and multiple neural network architectures for non-linear systems. Examples for the latter are locally recurrent locally feedforward networks (LRGF), Multi Layer Perceptrons (MLP) and Radial Basis Function (RBF) networks of different types of dynamics. These model structures are generalized, however, and as we will show, further theoretical background on linear systems theory could be leveraged.

Generally, deep learning offers multiple neural network layer types that can be employed when dealing with sequence-to-sequence problems, like fully connected (FC) networks, convolutional networks (CNN) or recurrent networks. Recurrent networks are also known as sequential models (like RNN, LSTM by Hochreiter & Schmidhuber (1997) and GRU by Cho et al. (2014)) and have been used successfully for text based sequence-to-sequence problems like machine translation or text processing. Wang (2017) demonstrates a concept of LSTM for dynamic system identification by using several parallel LSTM layers which predict the systems behaviour based on its input and prior predictions and their derivatives. A different approach of modelling dynamic systems are neural ordinary differential equations (ODEs) by Chen et al. (2018). These networks learn $dy/dt$ of a function $f$ with $y(t) = f(x(t))$ and the resulting ODE model is used an numerical integrator/solver (like Runge-Kutta) to compute $y(t)$. This has the advantage of a varying sampling step size which is determined by the solver, but these methods are agnostic of dynamic systems theory knowledge. Similarly, Raissi et al. (2019) use deep learning to learn Partial Differential Equations (PDE) of physical systems in a FC model combined with a numerical integrator. Furthermore, since the evaluation of ODE/PDE models is done using a numerical integrator, the model is difficult to apply in combination with other neural network layers like for example convolutional or recurrent layers. In terms of sampling frequency of the measurement data, recurrent network architectures can only be trained on one specific data frequency, and do not provide the functionality to generalize to other sampling rates of the same system. In such a case one would have to resample the new data to the frequency of the training set.

Explainability approaches for sequential models in text processing deduce which parts of the sentences are relevant for the model's prediction based on the activation of the internal gates (as shown by e.g.Krakovna & Doshi-Velez (2016)). Interpretability of RNN/LSTM or GRU models for continuous measurement data has not been explored yet to our knowledge.

## 3 DYNAMIC RECURRENT NETWORK

The complexity of the modelling of dynamic systems does not only result from the potential non-linearity, but also from the fact that the model has to keep track of the system's current and past states in order to predict the output based on new input. We intend to model a dynamic system in

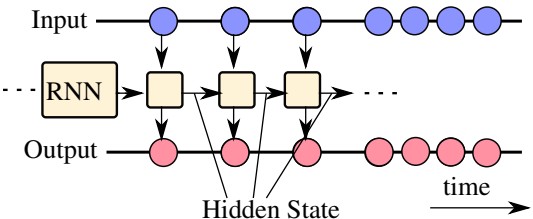

Figure 2: Example of sequence-to-sequence translation using a recurrent network layer. The layers hidden state is used for the computation of the next timestep's result.

a sequence-to-sequence task, which is why we chose a recurrent network architecture. Recurrent neural networks can be seen as units which iterate over a given sequence and predict an output. During this computation, a hidden state is computed which is leveraged for the prediction of the next time step. Our network differs from other state of the art recurrent networks in its computation function, which is derived from linear dynamic systems theory.

In the following we explain the theoretical background used in this work. Then we describe the structure of our DYRNN. Finally, we show different advantages that result from this structure like training and prediction at different signal sampling rates and interpretability of the models.

### 3.1 LAYER STRUCTURE

The background knowledge in this section is covered by Föllinger et al. (2016) and Yarlagadda (2010). In dynamic systems theory a linear system with the external input $u(t)$ and resulting system output $y(t)$ is expressed as the differential equation

$$y(t) + a_1 \cdot \dot{y}(t) + \ldots + a_n \cdot y^{(n)}(t) = b_0 \cdot u(t) + b_1 \cdot \dot{u}(t) + \ldots + b_n \cdot u^{(n)}(t). \quad (1)$$

Therefore, a linear system acts as transformation of the input $u(t)$ based to a convolution ($*$) with the system's linear transfer function $g(t)$ to produce the system output $y(t)$ with

$$y(t) = u(t) * g(t) = \int u(\tau)g(t-\tau)d\tau. \quad (2)$$

The transfer function $g(t)$ is used in engineering for, amongst others, controller design, system stability analysis or frequency response estimation. Larger, more complicated transfer functions can be seen as several basic functions which are interconnected in a circuit type fashion in parallel or in series (see Figure 3).

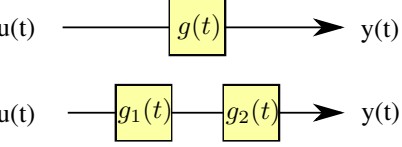

Figure 3: Examples for transfer function circuits. Linear systems are modeled by chaining together several transfer functions.

Dynamic systems theory identifies five basic linear transfer functions from which all linear dynamic systems can be modeled, which we use in the construction of the DYRNN: P, I, D, PT1 and PD. For further information and and visualization see Appendix Section A). They have the following functionalities:

- P: **P**roportional gain of the input, implemented as a multiplication with a constant
- I: **I**ntegrating component, meaning the step-wise summation of an input over time.
- D: **D**ifferential component acting as a high-pass filter

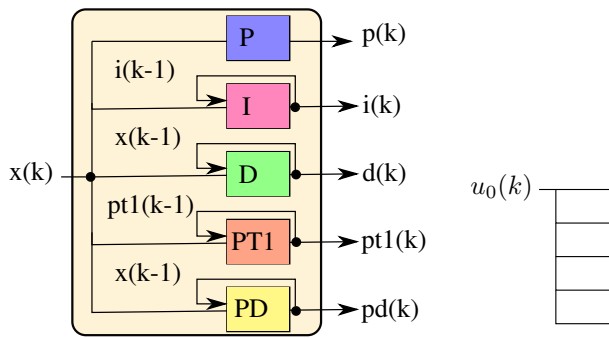 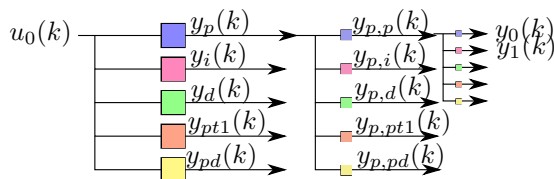

(a) Design of DYRNN unit with five different inner equations. This architecture results in multiples of five output channels.

(b) Resulting model when stacking DYRNN layers. Each of the output channels would produce another five output channels for each connected sub-component.

Figure 4: Structure of DYRNN and how appending layers is performed.

- PT1: **P**roportional multiplication of input with **t**ime delay, used to model e.g. voltage in capacitors in RC circuits. This function also acts as a low-pass filter

- PD: **P**roportional increase with a **d**ifferential component

In the following equations K stands for a constant with influence on the output amplitude, while T is a time constant which influences the speed of the system's reaction. K and T act as the trainable weights in a DYRNN layer. For usage in a recurrent layer, these differential equations are discretized with – in our case – first degree forward Euler. This is common in many engineering applications, and means replacing $\dot{y}(t)$ with

$$\dot{y}(t) = \frac{y(k) - y(k-1)}{\Delta t}. \tag{3}$$

This results in discrete recurrence equations, with the sample number k and the time distance between samples $\Delta t$. The respective equations of the basic transfer functions are as follows:

$$p(k) = K_P \cdot x(k) \tag{4}$$

$$i(k) = i(k-1) + \frac{\Delta t}{K_I} \cdot x(k) \tag{5}$$

$$d(k) = \frac{K_D}{\Delta t} \cdot (x(k) - x(k-1)) \tag{6}$$

$$pt1(k) = pt1(k-1) + (K_{PT1} \cdot x(k) - pt1(k-1)) \cdot \frac{\Delta t}{\Delta t + T_{PT1}} \tag{7}$$

$$pd(k) = K_{PD} \cdot (x(k) + \frac{T_{PD}}{\Delta t} \cdot (x(k) - x(k-1))), \tag{8}$$

with the input $x(k)$ and all $K, T > 0$.

The equations above are implemented as the computation function of a recurrent network layer as described in Appendix Section A.1. K and T in the equations become trainable weights of the layer, while the hidden state consists of $x(k-1), i(k-1)$ and $pt(k-1)$. In the following we refer to these internal computations as subcomponents. We explore two different network variants in our work: The DYRNN5 with all five known subcomponents and the DYRNN3 with just P, PD and PT1. The reason for this is, that a D subcomponent can be approximated by PD and I by a PT. Since integrators can also cause model instabilites, we explore both variants in our experiments.

We formulate one unit's computations based on the number of input channels $n_{ic}$ and the number of outputs per subcomponent type $n_{oc}$. The number of output channels per layer $n_{layer}$ amounts to (subcomponent count)* $n_{oc} = 3 * n_{oc}$ for DYRNN3 and 5*$n_{oc}$ for DYRNN5 (see Figure 4a).

The fact that DYRNN can be implemented with an interface compatible with other recurrent networks allows the modelling of a systems properties by training in a sequence-to-sequence fashion

using out of the box backpropagation through time, gradient descent and optimizer functions as implemented in state of the art frameworks like Pytorch (by Paszke et al. (2019)) and Tensorflow (by Martín Abadi et al. (2015)).

Since each input of the layer is connected to five basic components, stacking DYRNN layers results in an increasing number of output channels, as shown in Figure 4b. In our experiments, we achieved good results with two cascading DYRNN layers followed by a per-time step linear FC layer with one neuron. This final layer can be enhanced using non-linear activation functions and more layers/neurons, which would result in a similar structure to Hammerstein-Wiener models for modelling of static non-linearities in the data. In case of noisy input data, for example due to sensor noise, additional CNN layers could be used as well.

## 3.2 GENERALIZATION FOR DIFFERENT SIGNAL SAMPLING RATES

The new parameter $\Delta t$ means that a model can be used on datasets with a different sampling rates by adjusting the $\Delta t$ when interacting with the model. This can be leveraged as a new means to perform time series data augmentation. Training on different rates and testing on another sampling rate can also provide insight about a model's robustness. Additionally, training time can be reduced by subsampling the dataset during training, and using the original sampling rate while inferencing for greater prediction accuracy (see Figure 5).

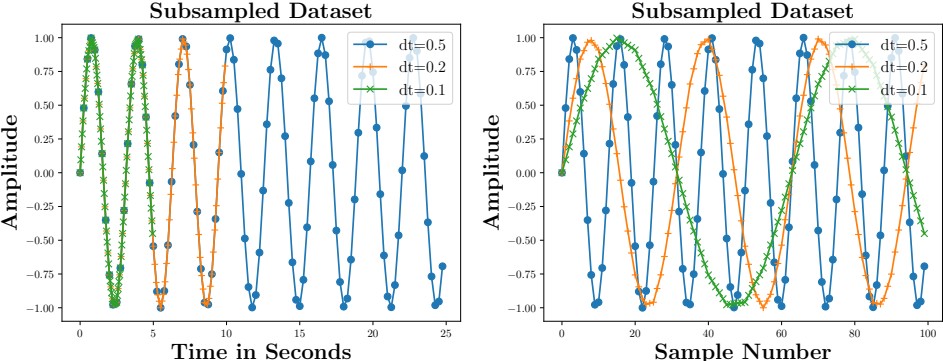

Figure 5: Example dataset y(k) = sin(x(k)) at different sampling rates. Keeping the amount of samples the same, a larger sampling rate results in a larger range of covered time.

## 3.3 INTERPRETABILITY OF RESULTING MODELS

Based on our layer, it is possible to interpret the significance of a subcomponent towards the final dataset as well as the properties of its influence. While the DYRNN layers learn the time dependant system behaviour, a per-timestep fully connected network selects the relevant dynamics of the model. The higher the FC's learnt weight connected to a specific subunit, the more significant it is for the modelling of the dynamic system (see Figure 6). As an example, assume a network of one DYRNN5 layer and a fully connected (FC) layer as shown in Figure 6. Since the subcomponents P, I and PT1 are weighted with 0.5, 1.5 and 0.8 as opposed to the D and PD with 0.1 and 0.2, the modeled system mostly displays P, I and PT1 properties.

Additionally, we can analyse the model by evaluating its equation in the Laplace and the Fourier domain. This means replacing the discrete subcomponents with their respective Laplace formulas as described in the Appendix Section A, while keeping the structure of the network as well as the trained values for K and T. This yields a new equation depending on the parameter $s$, which for the example network above results in:

$$Y(s) = U(s)(0.5 + \frac{1.5K_I}{s} + 0.1sK_D + \frac{0.8K_{PT}}{(1 + sT_{PT})} + 0.2K_{PD}(1 + sT_{PD})) \tag{9}$$

$$Y(s) = U(s)\frac{0.1s^3K_DT_{PT} + s^2(0.1K_D + ...) + s(1.5K_IT_{PT} + ...) + 1.5K_I}{1.0s^2T_{PT} + 1.0s} \tag{10}$$

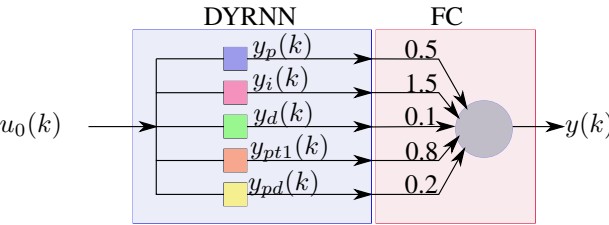

Figure 6: Interaction of one DYRNN layer and a FC network layer. Via the FC layer's learnt weights, it is possible to gain insight of the underlying system dynamics.

The Laplace form has the advantage that a convolution with the transfer function $y(t) = u(t) * g(t)$ can be replaced with a multiplication with $Y(s) = G(s) \cdot U(s)$. It allows an engineer to analyse the model for stability, and to visualize different kinds of information on the model, e.g. in a pole-zero plot. The transfer function $G(s)$ can be examined for system stability by replacing s with $i\omega$ ($\omega = 2\pi * f$, with $f$ as the signal frequency) yields the Fourier from of the transfer function. This can be used for frequency response and root locus analysis (Föllinger et al. (2016)), as we show in our experiments.

The transformation of $G(s)$ for our example network into the time domain yields the following differential equation:

$$y''(t)T_{PT} + y'(t) = u'''(t)0.1K_D T_{PT} + u''(t)(0.1K_D + \cdots) + u'(t)(1.5K_I T_{PT} + \cdots) + u(t)1.5K_I \tag{11}$$

Transforming larger networks – for example the one used in our experiments – works along the same lines. Such extracted differential equations can further be leveraged to analyze the model.

## 4 EXPERIMENTS

We evaluated our network on three different datasets. The first is an electrical RC-circuit simulated using Matlab Simulink, as shown in the Appendix in Figure 13. The other two datasets are from the Database for Identification of Systems "DaISy", namely the Heating System De Moor B.L.R. (b) and the Thermic Resistance De Moor B.L.R. (a)) datasets. In the following we will refer to the datasets as Circuit, Heating and Thermic. In these experiments we focus on learning the correct dynamics independent from the initial value of y(t). Therefore, we subtract the mean time offset of the prediction sequence towards the label before the final evaluation with Mean Squared Error.

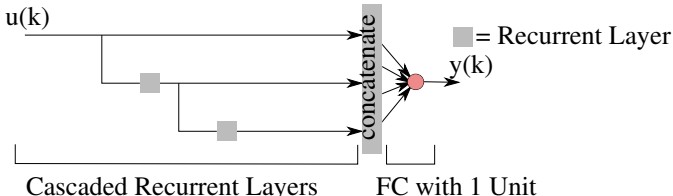

Figure 7: Architecture used in our experiments.

The first goal of the experiments is to evaluate our models (DYRNN3 and DYRNN5) compared to RNN, LSTM and GRU as implemented in Tensorflow. The network architecture is shown in Figure 7. The first part of the network is built in a cascaded fashion of two recurrent layers which are concatenated with the original input. Its results are passed to a linear fully-connected layer with one unit. The number of units in the other recurrent layers were chosen to result in a similar parameter distribution to DYRNN5, as shown in Section A.2. For experiment a similar count of parameters in the recurrent part of the network and the same layer constellations were used (see Table 4,5 and 6). Due to the design of the DYRNN5 network, its FC part contains more parameters than the other networks.

Additionally, a DYRNN5 model was trained for different sampling rates for the Circuit dataset to examine its ability to generalize to other sampling rates. For this, the dataset was subsampled at different frequencies using linear interpolation. The constellation of sampling rates in these three runs are described in Table 1. All networks were trained with the Adam Optimizer by Kingma & Ba (2014) and the Huber Loss function with standard parameters defined in Tensorflow. The amount of training iterations was 3000 epochs. Evaluation of results is done on a separate testing dataset.

Table 1: Amount of parameters per network layer and recurrent layer type.

| Run | Training set resampled at | Prediction set resampled at |
|-----|---------------------------|------------------------------|
| A | $0.7 \, \Delta t_{orig}$ | $1.0 \, \Delta t_{orig}$ |
| B | $1.0 \, \Delta t_{orig}$ | $0.7 \, \Delta t_{orig}$ |
| C | $0.5 \, \Delta t_{orig}, 1.0 \, \Delta t_{orig}$ | $0.7 \, \Delta t_{orig}$ |

After training the DYRNN models are transformed into Laplace transfer functions G(s) using Sympy (Meurer et al. (2017)) and three different views of this function are generated: G(s) over s, the frequency response $|G(i\omega)|$ over $\omega$ and the root locus curve of $G(i\omega)$ over $\omega$ to compare the different model dynamics per dataset.

## 5 RESULTS

After 10 runs each, the different recurrent networks' performance on the testing set is shown in Figure 8. Since an activation function in the FC layer was omitted, it amounts to a linear combination of all channels from the cascaded part. Therefore, it can be assumed that a better performance of the DYRNN stems from its computation functions. In this case a different network configuration and more parameters may enable the other layer types to achieve better results.

Section A.3 in the Appendix shows the best and worst runs on the testing set for all datasets and layer types as well as the transfer functions learnt by DYRNN3 rounded to two significant digits in Section A.4. The transfer function analysis plots in Figures 10, 11 and 12 were generated for the runs which performed best on the testing datasets.

On the Circuit dataset, both DYRNN5 and DYRNN3 perform consistently better than each of the other recurrent networks. This is to be expected since an idealized capacitor acts as a PT1 element, so the DYRNN is predestined for this modelling task. Nonetheless, all layer types perform well on the dataset if training was successful. The RNN network, for example, was unable to fit the data in a meaningful way on some runs (Appendix Figure 14). Figure 10 shows that G(s) for DYRNN5 has more poles. This is to be expected because of the structure of the network, but can be an indicator for instability in some frequency areas. G(s) of DYRNN3 follows the given plots better. The Frequency responses of both DYRNN networks are similar to the actual system, but the root locus curve of DYRNN5 differs significantly from the actual one: its starting point is far outside of the plotted region.

On the Heating dataset, the GRU network performed slightly better than DYRNN3 and DYRNN5. It is noticeable that, while the datasets is visually similar to a PT1 element, the DYRNN3 model displays a similar frequency response plot and root locus curve in Figure 11 b) and c) as seen in the actual model in the PT1 system in the Circuit dataset (Figure 10).

On the Thermic dataset, DYRNN3 performed similarly to GRU. The dataset consists of two inputs and one output, yielding two transfer functions $G_1(s)$ and $G_2(s)$. These can be analysed separately as shown in Figure 12 for DYRNN3. While $G(s)$ over $s$ is similar to the one in Circuit, the frequency response and root locus of both $G_1(s)$ and $G_2(s)$ are very different from Circuit and Heating.

In total, DYRNN5 performed worse than DYRNN3 on the non-simulated datasets. We assume that the reason for this is that Heating and Thermic do not have a strict zero-level at the start and the end of the datasets' input and output. That might enforce a more stable model.

Our results concerning resampling to different sampling rates (in 9) show that training for $0.7 \, \Delta t$ and predicting to $1.0 \, \Delta t$ performs worse than the other two variants. From this, it can be concluded that

the total amount of time covered by the training data is more important than the training at different sampling rates for this dataset. It also shows that the model is able to extrapolate for each of the different sampling rates to another. Our implementation, which allows the per-batch sampling rate during training is computationally more expensive than the one which relies upon a fixed sampling rate, so the inexpensive version is a possible alternative for larger datasets.

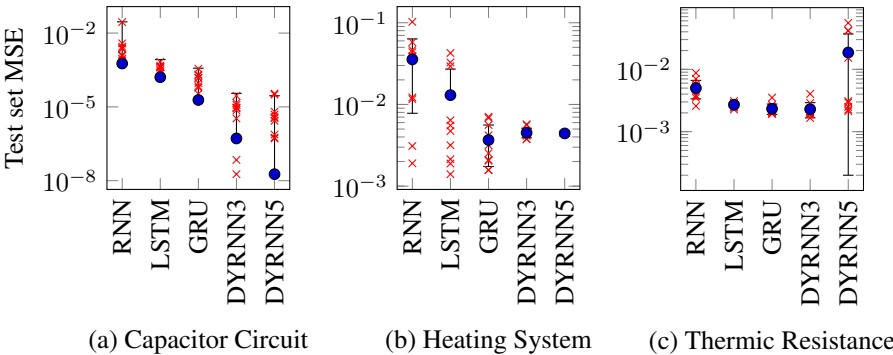

(a) Capacitor Circuit     (b) Heating System     (c) Thermic Resistance

Figure 8: Results on different evaluation datasets after 10 runs each using the same architecture with varying layer types.

Table 2: MSE and Standard deviation of the different layer types on the experiment datasets.

|  | RNN | LSTM | GRU | DYRNN3 | DYRNN5 |
|---|---|---|---|---|---|
| Circuit | 4.4e-3±7.6e-3 | 3.8e-4 ±1.3e-4 | 1.3e-4 ± 9.2e-5 | 9.3e-6 ±7.4e-6 | **9.0e-6± 1.2e-5** |
| Heating | 3.6e-2 ± 2.8e-2 | 1.3e-2 ± 1.4e-2 | **3.7e-3± 1.9e-3** | 4.5e-3 ± 6.0e-4 | 4.4e-3± 3.1e-4 |
| Thermic | 5.0e-3 ± 1.7e-3 | 2.7e-3 ±2.2e-4 | 2.3e-3± 4.4e-4 | **2.3e-3± 6.3e-4** | 1.9e-2± 1.8e-2 |

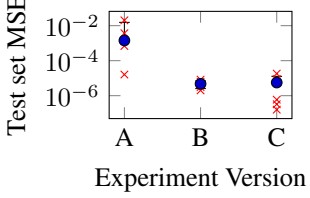

Experiment Version

|  | A | B | C |
|---|---|---|---|
| Mean Loss | 1.43e-3 | **4.76e-06** | 5.63e-06 |
| Loss Std.-dev. | 1.4e-3 | 2.21e-06 | 7.11e-06 |

Figure 9: Results on the test set. A: $0.7\Delta t_{orig} \rightarrow 1.0\Delta t_{orig}$; B: $1.0\Delta t_{orig} \rightarrow 0.7\Delta t_{orig}$; C: $0.5\Delta t_{orig}, 1.0\Delta t_{orig} \rightarrow 0.7\Delta t_{orig}$

## 6   FUTURE WORK

We have shown a new type of recurrent layer designed for the modelling of dynamic systems. Based on this, we see several potential areas for future work. One point that was not part of this work was the in depth analysis of the learnt transfer functions, and how the transfer function is to be interpreted in case of non-linear activation functions in the network. The plots shown of the transfer functions are accessible and interpretable mainly for engineers. Another area of research would be on how to make these results interpretable for scientists without an engineering background.

Our experiment showed competitive results on three datasets of system identification. A different area of application can be model based reinforcement learning tasks, since the layers' computation blocks are also commonly used in control engineering.

## 7   CONCLUDING REMARKS

In this work we present a new recurrent network architecture which is based on dynamic systems theory. We show that the learnt system dynamics can produce models which can extrapolate to different sampling rates in the data. Due to the specific meaning of the cells' computation function, the

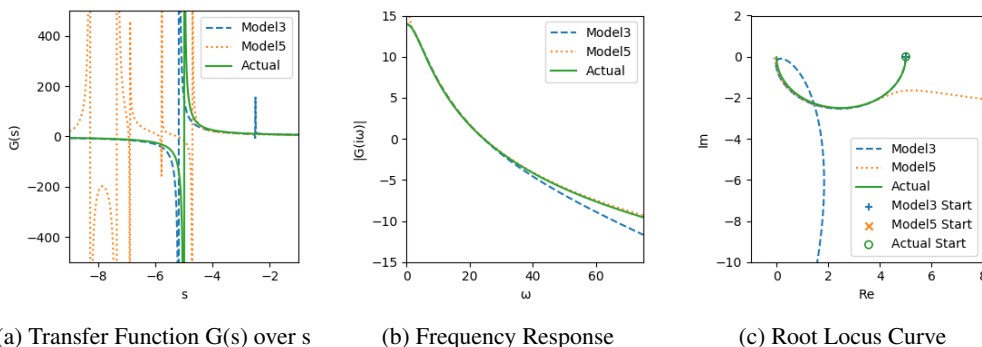

(a) Transfer Function G(s) over s    (b) Frequency Response    (c) Root Locus Curve

Figure 10: Circuit model transfer function. Actual data could be computed since the resistance and capacitance of the circuit are known.

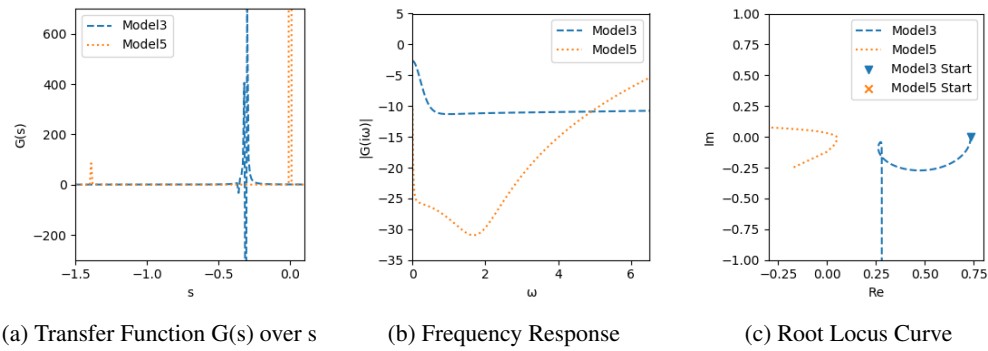

(a) Transfer Function G(s) over s    (b) Frequency Response    (c) Root Locus Curve

Figure 11: Heating system transfer function for DYRNN3 and DYRNN5. Even though the models are of similar performance, differences can be seen using this advanced visualization.

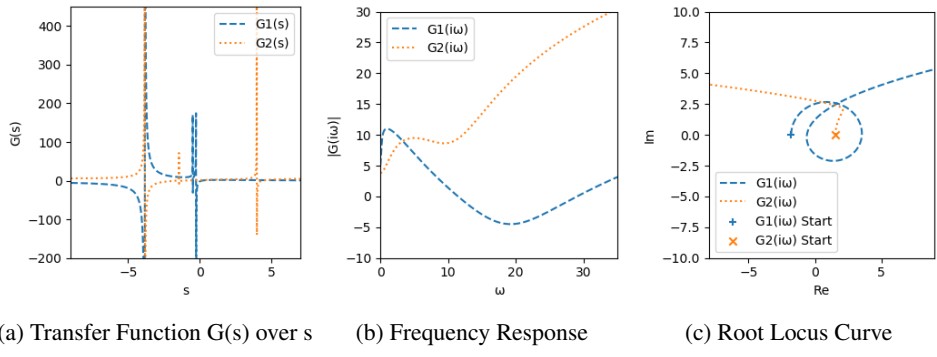

(a) Transfer Function G(s) over s    (b) Frequency Response    (c) Root Locus Curve

Figure 12: DYRNN3 results on the Thermic dataset. Since the system has two different inputs, the model provides one transfer function each.

learnt model can be leveraged to gain insight on the underlying system dynamics and a differential equation model of the system can be extracted.

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

# A APPENDIX

BASE FUNCTIONS IN DYNAMICAL SYSTEMS THEORY

In dynamic systems theory, the basic elements are visualized using their response towards an input unit step function $\sigma(t)$ starting at t=0 with the amplitude of 1. In Table 3, we show a short summary over the blocks used in our network layer. Blocks display a specific behaviour, which is visualized by looking at the response towards the Unit step function.

## A.1 IMPLEMENTATION DETAILS

Sequential models keep a hidden state while iterating over an input sequence, to encode information necessary for the computation of the next step's result.
There are two ways to implement the model: one where the sampling rate is kept constant between batches and one where the sampling rate can be set per batch. While the first version can be implemented using trivial matrix multiplication, the per-batch version is implemented as follows:

$$p(k) = u(k) \circ |K_P| \tag{22}$$

$$i(k) = i(k-1) + \left(i(k) \bullet \frac{1}{\Delta t}\right) \circ \frac{1}{|K_I|} \tag{23}$$

$$d(k) = (u(k) - u(k-1)) \bullet \frac{1}{\Delta t} \circ |K_D| \tag{24}$$

$$pt1(k) = pt1(k-1) + (u(k) \circ |K_{PT1}| - pt1(k-1)) * \frac{1}{\frac{1}{\Delta t} \circ |T_{PT1}| + 1} \tag{25}$$

$$pd(k) = \left(u(k) + ((u(k) - u(k-1)) \bullet \frac{1}{\Delta t}) * |T_{PD}|\right) \circ |K_{PD}|. \tag{26}$$

$$\tag{27}$$

With $u(t)$'s dimensionality of [batch × sample × input channels] and $\Delta t(t)$ as [batch × sample × 1] or $\Delta t$ as [batch × 1 × 1] , the multiplications expressed in Einstein's sum formation (which can be implemented as described in e.g. https://www.tensorflow.org/api_docs/python/tf/einsum ) are: ∘ as 'bi, ij → bij', • as 'bi, b → bi' and ∗ as 'bij, bij→bij', i.e. the Hadamard product operator. The dimensions of the trainable parameters result to

$$K_P \in \mathbb{R}^{ic \times oc}$$
$$K_I \in \mathbb{R}^{ic \times oc}$$
$$K_D \in \mathbb{R}^{ic \times oc}$$
$$K_{PT1} \in \mathbb{R}^{ic \times oc}, T_{PT1} \in \mathbb{R}^{ic \times oc}$$
$$K_{PD} \in \mathbb{R}^{ic \times oc}, T_{PD} \in \mathbb{R}^{ic}$$

given this setup. The weights of P,D,PT1 and PD were initialized in a random uniform distribution, with a range $[0.1, 0.2]$. Its important to initialize I such that the first iteration does not become unstable. For datasets with around 500 elements, the initialization used in the experiments was in the range of $[110.0\Delta t, 110.0\Delta t]$.

## A.2 FURTHER EXPERIMENT DOCUMENTATION

We evaluated our network on three different datasets. One is our own dataset from a Matlab (MATLAB (2018)) simulation as shown in Figure 13. The input of our model is the input voltage of the

Table 3: Basic linear components, input $\sigma(t)$ ····· and system response $y(t)$ ——. Step response and differential equations of the used linear components.

| System | Unit Step Function Response | General Block Output Function | Laplace Form |
|---|---|---|---|
| P |  | $y(t) = K \cdot u(t) \qquad (12)$ | $Y(s) = K \cdot U(s) \qquad (13)$ |
| I |  | $\dot{y}(t) = K \cdot u(t) \qquad (14)$ | $Y(S) = U(s)/s \qquad (15)$ |
| D |  | $y(t) = K \cdot \dot{u}(t) \qquad (16)$ | $Y(s) = U(s) \cdot s \qquad (17)$ |
| PT1 |  | $T \cdot \dot{y}(t) + y(t) = K \cdot u(t) \qquad (18)$ | $Y(s) = U(s)\dfrac{K}{(1 + T \cdot s)} \qquad (19)$ |
| PD |  | $y(t) = K \cdot (\dot{u}(t) \cdot T + u(t)) \qquad (20)$ | $Y(s) = U(s) \cdot K \cdot (1 + T \cdot s) \qquad (21)$ |

circuit, while the output value is the voltage measured over the capacitor. The simulation's sampling rate is 0.005s. We simulated three time series for training, validation and testing with length of 2 seconds. The input consists of a square signal with randomly changing amplitudes from 0.5s to 1.5s. The remainder of the time, the input voltage is 0V.

The other two are the "Heat flow density through a two layer wall" (De Moor B.L.R. (a)) (in short Thermic) and the "Heating system" (De Moor B.L.R. (b)) (in short Heating) datasets from the DaISy system identification benchmarks:

- Heating System Benchmark: Prediction of halogen lamp temperature based on input voltage

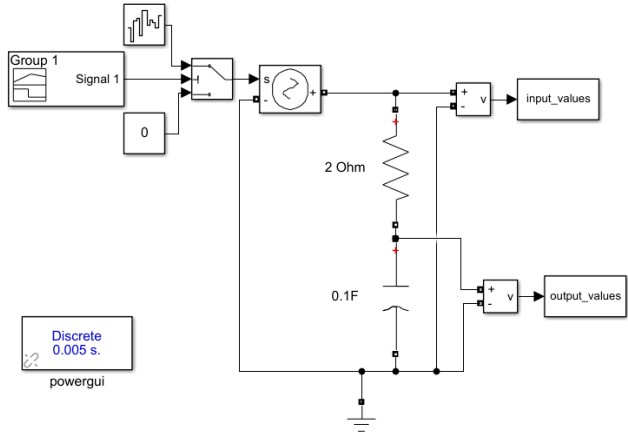

Figure 13: Simulation used for the generation of experiment data. It contains a resistor of 2 $\Omega$ and a capacitor of 0.1F. Input sequence is the source voltage, while output to predict is the voltage measured over the capacitor.

- Two Walls Benchmark: Prediction of heat flow through two walls based on measurements before and between the walls

After training, the models were evaluated on a testing set split from the complete dataset. The DaISy Datasets do not have a zero-level starting interval like in our circuit dataset. For simplicity's sake, our chosen network architecture does not take the initial value of the output time series into account. Therefore an offset could be seen in predictions on the testing set for all layer types. In a post processing step this offset was computed using the median difference between prediction and label data, and subtracted from the prediction prior to the final MSE evaluation.

The amount of units and parameters for all experiments are shown in the Tables 4, 5, 6.

Table 4: Amount of units per network layer and recurrent layer type for the the Circuit dataset. Values in parentheses denote the parameter count of that layer.

| Units (Param. Count) | RNN | LSTM | GRU | DYRNN5 | DYRNN3 |
|---|---|---|---|---|---|
| Units in first recurrent layer | 3 (15) | 2 (32) | 2 (30) | 1 output per s.c. (7) | 1 output per s.c. (5) |
| Units in second recurrent layer | 5 (45) | 2 (40) | 2 (36) | 1 output per s.c. (35) | 1 output per s.c. (15) |
| Number of parameters in FC layer | 10 | 6 | 6 | 32 | 14 |
| Number of parameters total | 70 | 78 | 72 | 74 | 34 |

Table 5: Amount of units per network layer and recurrent layer type for the Heating System dataset.

| Units (Param. Count) | RNN | LSTM | GRU | DYRNN5 | DYRNN3 |
|---|---|---|---|---|---|
| Units in first recurrent layer | 3 (15) | 2 (32) | 2 (30) | 2 output per s.u. (14) | 1 output per s.u. (5) |
| Units in second recurrent layer | 7 (77) | 3 (72) | 3 (63) | 1 output per s.u. (70) | 1 output per s.u. (15) |
| Number of parameters in FC layer | 12 | 7 | 7 | 63 | 14 |
| Number of parameters total | 104 | 111 | 100 | 147 | 34 |

Table 6: Amount of units per network layer and recurrent layer type for the Thermic Wall Res. dataset.

| Units (Param. Count) | RNN | LSTM | GRU | DYRNN5 | DYRNN3 |
|---|---|---|---|---|---|
| Units in first recurrent layer | 3 (18) | 2 (40) | 2 (36) | 1 output per s.u. (14) | 1 output per s.u. (10) |
| Units in second recurrent layer | 7 (77) | 3 (72) | 3 (63) | 1 output per s.u. (70) | 1 output per s.u. (30) |
| Number of parameters in FC layer | 13 | 8 | 8 | 63 | 27 |
| Number of parameters total | 108 | 120 | 107 | 147 | 67 |

### A.3 BEST AND WORST RESULTS ON DATASETS

The Figures 14, 15 and 16 show the best and worst testing results of all 10 runs per network type.

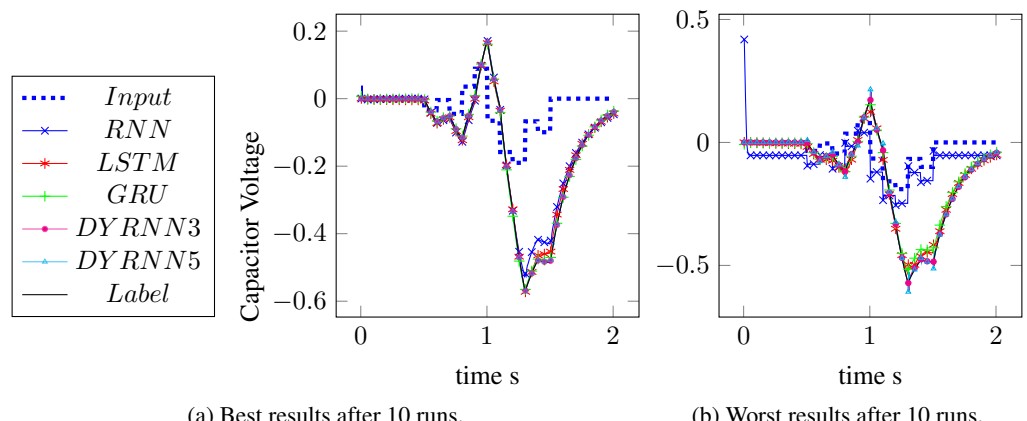

(a) Best results after 10 runs.  (b) Worst results after 10 runs.

Figure 14: Comparison of best and worst models on the Circuit dataset.

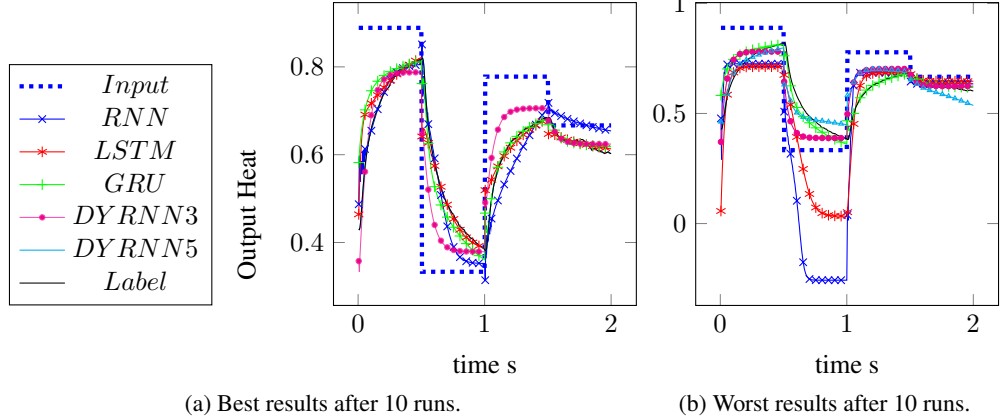

(a) Best results after 10 runs.  (b) Worst results after 10 runs.

Figure 15: Comparison of best and worst models on the Heating dataset.

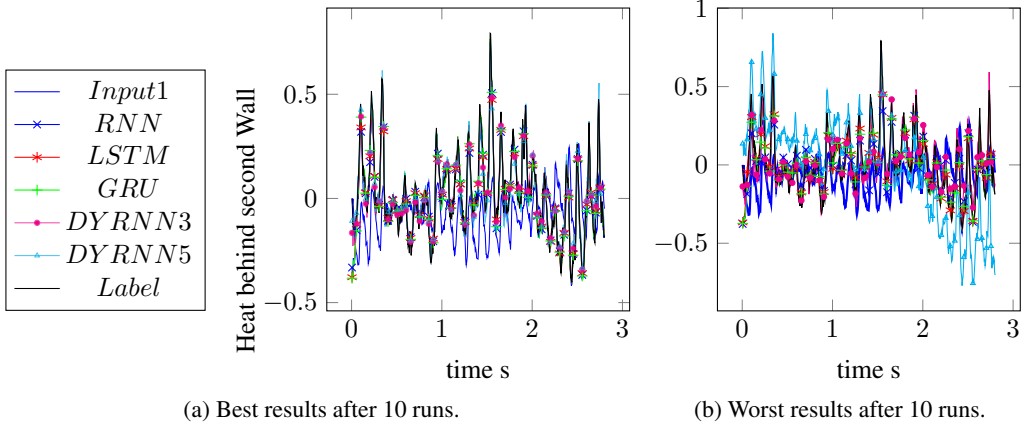

(a) Best results after 10 runs.     (b) Worst results after 10 runs.

Figure 16: Comparison of best and worst models on the Thermic dataset.

## A.4 RESULTING TRANSFER FUNCTIONS

Below the transfer functions of the best runs are documented. The transformation to differential equations is trivial. Notice that due to the rounding to 2 significant digits, the differential equations listed here are unlikely to yield accurate results. Additionally, simplification of these transfer functions is possible if the poles and zeros of the function match, but this is outside of the scope of this work. The transfer functions of the best runs of DYRNN3 are as follows:

Circuit:

$$G(s) = \frac{9.2E-10s^6 - 3.2E-7s^5 + 3.6E-5s^4 - 0.0021s^3 + 0.75s^2 + 3.9s + 5.0}{1.6e-5s^4 + 0.03s^3 + 0.3s^2 + 0.98s + 1.0} \tag{28}$$

Heating:

$$G(s) = \frac{-2.9E-9s^7 - 0.04s^6 + 0.85s^5 + 3.8s^4 + 4.5s^3 + 2.5s^2 + 0.68s + 0.073}{3.6s^5 + 14.0s^4 + 14.0s^3 + 6.2s^2 + 1.3s + 0.099} \tag{29}$$

Thermic Resistance:

$$G(s) = G_1(s) + G_2(s) \tag{30}$$
$$Y_1(s) = X_1(s) \cdot G_1(s) \tag{31}$$
$$Y_2(s) = X_2(s) \cdot G_2(s) \tag{32}$$
$$G_1(s) = \frac{-0.0025s^6 + 0.12s^5 - 2.1s^4 + 31.0s^3 + 40.0s^2 + 7.9s - 1.8}{2.3s^4 + 13.0s^3 + 16.0s^2 + 7.3s + 1.0} \tag{33}$$
$$G_2(s) = \frac{2.2E-8s^6 + 0.0013s^5 + 0.01s^4 + 0.15s^3 - 0.4s^2 - 2.0s - 1.5}{7.6E-7s^4 + 0.045s^3 + 0.06s^2 - 0.69s - 1.0} \tag{34}$$

This sum of transfer functions can be used to simulate $y(t)$ by first simulating $y_1(t)$ and $y_2(t)$ separately and then building the sum of the results.

