# OpenReview forum: "Recurrent Neural Network Architecture based on Dynamic Systems Theory for Data Driven Modelling of Complex Physical Systems"
_ICLR.cc/2021/Conference — Reject_

### Official Review · AnonReviewer3 · 2020-10-24
**Clear reject. Lacks clarity. Findings not likely to be very general (weak significance).**

**Rating:** 3
**Confidence:** 5

**Review:**

The authors present a method for incorporating basic concepts from linear systems theory into the standard structure for training artificial neural networks. They compare results of their approach against standard approaches for 3 simple datasets.

Broadly speaking the work is quite unclear, and takes several passes over to have a basic sense of the approach. There are too many shortcomings to enumerate them all, so I will just present one. Figure 5 is presented in the section "example architecture" which might lead one to believe the authors implement this network (which it appears they do not). I believe this is included only to indicate a hypothetical architecture, but the presentation is too poor to glean this with any quickness. This is of course, in itself, not sufficient grounds for rejection, but speaks broadly to the poor presentation of the work. It does not seem ready for publication.

As for the significance, the work clearly falls short. Although the motivation of constructing a "more explainable model" is a good one, this should not come at an extreme cost of model expressivity. It seems obvious that richer models, such as LSTMs etc., correctly trained, should be able to account for the linear transformations the authors include in their "novel layer." That their work is competitive with these richer models is simply an indication of the simplicity of the tasks they chose, which (as far as I can tell) can all be accounted for using linear systems analysis (although it's hard to say, since they work so poorly explains the second two tasks). It's completely unclear how effective the authors' approach would be over standard, richer models, on tasks that cannot be accounted for by linear systems analysis, and I am doubtful that the suggested approach could offer much over these richer models.

Likewise, an alternative view of the authors' work is as a learnable filter bank applied to data to create a representation of the data better suited for post-hoc learning with a richer model, which is certainly an useful idea, but it is not clear to me (and the authors haven't shown) that their choice for this filter-bank is superior to many other choices (e.g. convolutional layers applied prior to FC layers, which is standard for deep networks).

---

> ### Author Response · Authors · 2020-11-23
> **Response to AnonReviewer3**
>
> Thank you very much for reading our work and for your feedback. We are sorry that you found the paper difficult to follow and have revised large parts of it in an effort to improve this issue.
>
> > Broadly speaking the work is quite unclear, and takes several passes over to have a basic sense of the approach. There are too many shortcomings to enumerate them all, so I will just present one. Figure 5 is presented in the section "example architecture" which might lead one to believe the authors implement this network (which it appears they do not). I believe this is included only to indicate a hypothetical architecture, but the presentation is too poor to glean this with any quickness.
>
> This was an error in the text (around Figure 5) and the section been rewritten now, thank you for the pointer.
>
> > As for the significance, the work clearly falls short. Although the motivation of constructing a "more explainable model" is a good one, this should not come at an extreme cost of model expressivity. It seems obvious that richer models, such as LSTMs etc., correctly trained, should be able to account for the linear transformations the authors include in their "novel layer." That their work is competitive with these richer models is simply an indication of the simplicity of the tasks they chose, which (as far as I can tell) can all be accounted for using linear systems analysis (although it's hard to say, since they work so poorly explains the second two tasks). It's completely unclear how effective the authors' approach would be over standard, richer models, on tasks that cannot be accounted for by linear systems analysis, and I am doubtful that the suggested approach could offer much over these richer models.
>
> This is a valid point. Firstly, we added plots of the training datasets in order to show the  datasets used and rewrote the problem descriptions of both. In machine learning -- as we all know -- we often deal with the  trade-off between model expressivity and interpretability. We know that in non-linear problems non-linear models might be necessary. Nonetheless, many real-life systems can be linearized in their operation points and therefore be modelled adequately using linear models. Linear models are (most of the time) easier to understand for humans. Furthermore, our model does not rule out non-linear activation functions, they were just not the focus of this publication.
>
> > Likewise, an alternative view of the authors' work is as a learnable filter bank applied to data to create a representation of the data better suited for post-hoc learning with a richer model, which is certainly an useful idea, but it is not clear to me (and the authors haven't shown) that their choice for this filter-bank is superior to many other choices (e.g. convolutional layers applied prior to FC layers, which is standard for deep networks).
>
> We understand that there are different types of neural networks applicable for such tasks.
> For dynamic systems modelling, it is necessary to take the current and past state of the system into account.
> We did not use FC or CNN because to our knowledge, these models would only work in the following constellations:
> 1) Modelling of a PDE/ODE and using a separate ODE solver to compute the output sequence
> 2) One-step-ahead prediction, which can be computationally slow and cumbersome to implement
> 3) Sequence-to-sequence: impractical if one deals with measurement signals of changing lengths
> 4) Fully Convolutional sequence-to-sequence: while applicable to different lengths, the field of view of a convolutional kernel is limited and the kernel lacks the memory component of a recurrent network. Memory is needed e.g. for the modelling of an integrator
>
> We have expanded the interpretability section (3.3) and added new evaluation plots our experiments where we further analyse the model that stems from our approach. We hope that it will clarify our vision on why we created this work.

---

### Official Review · AnonReviewer1 · 2020-10-27
**overall this is a good submission**

**Rating:** 6
**Confidence:** 4

**Review:**

This paper aims at proposing Dynamic Recurrent Network to understand the underlying system properties of RNNs. By first showing five basic linear transfer functions in dynamic systems theory, the paper formulates DYRNN units. To solve the increasing number of layers issue, they concatenate inputs and intermediate results before passing into an FC layer. It is interesting to see how adjusting
\deta t is related to the model’s robustness. Though not fully explained, this paper provides a method to partially explain the RNN insights through FC layers learnt weights.

The paper is well written to convey the central ideas. The overall idea is interesting, and experiments are clear to demonstrate the proposed method. It will be better to test the method on some benchmark datasets so that it will be easy to compare with the state-of-the-art.

A small advice: do you mean varying instead of variing in section 6?

---

> ### Author Response · Authors · 2020-11-23
> **Response to AnonReviewer1**
>
>
> Thank you very much for reading our work and your kind feedback! We have revised our paper and have further explained the interpretability aspect of it. We will look into applying the model to more datasets in future work.
>
> We hope that you will like these changes and that they will clarify what we wanted to express regarding the interpretability of our model.

---

### Official Review · AnonReviewer4 · 2020-10-27
**Where’s the interpretability exactly?**

**Rating:** 4
**Confidence:** 5

**Review:**

The theory of dynamical systems has such a rich and extensive history dating to more than a century ago, that Poincaré himself would marvel to see what we can do now with modern technology.  Perhaps he’d not be pleased to see that extensive research in dynamical systems from the previous century is ignored in this paper, however the authors are aiming at reaching more interpretability on NN models as stated in the abstract.

I think the paper has definitely technical merits, but I remain puzzled with the lack interpretability section (3.4). I do not see how I can use the methodology to reconstruct the underlying physical processes that drive the physical phenomena. Since the authors are claiming in the abstract that interpretability is a distinctive feature of the methodology, I was expecting to see a full fleshed content, but it’s essentially just a couple of paragraphs.

These are the questions that I’d love the authors to address:
1/ Since the authors are inserting an explicit integration method by virtue of the Euler step through Eqs 3 to 8, the method inherently carries A-unstable. Can you estimate the conditions for any system that would prevent the A-stability risk?
2/ What are the differences of the methodology for Hamiltonian and dissipative dynamical systems?
3/ If the methodology that the authors are proposing works, it should be the case that any transient trajectory can be explained by DYRNN. It means that a trained DYRNN on a small set of trajectories (or one sufficiently large one if we have an ergodic system), can fully reproduce any other transient trajectory of the dynamical system. In other words, produce a train and test trajectories and prove that the error on the test set is small enough.
4/ At the core of classical dynamical system theory is understanding the set of local and global bifurcations that explain the physical/chemical/biological phenomena. If DYRNN can predict the bifurcation points respect to a control parameter, then the method is rock&roll, otherwise it’s just another method.
5/ Interpretability: If the authors can map back the DYRNN to a set of ODEs/PDEs then I’d be fully convinced on the value of the method. Otherwise, I remain unconvinced on the value of the interpretability.

I am not ready to accept this paper in its current form, but if the authors could prove (1,2) or 3 or 4 or 5 then I’ll happily change my rating.

---

> ### Author Response · Authors · 2020-11-23
> **Response to AnonReviewer4**
>
> Thank you very much for your kind and constructive review! It's exciting to have a reviewer who knows this much about the subject matter. Your questions are all very interesting and we would like to evaluate them in the future. Due to the limited time for this rebuttal, we have to postpone the answer for your points 1/ through 4/ to a later date and want to focus on your question about interpretability (5).
>
> We extended the according chapter (now 3.3) by explaining how one would first derive the transfer function of the network by transforming it into the Laplace domain while keeping the trained K and T weights from the subcomponents (P, I, D, PD and PT1). From there, one can derive not only the ODE of the system, but also perform different other analyses known from control engineering, like predicting the models' frequency response. We have also shown the according plots as part of our experiments, and attached example transfer functions for our experiment data in the appendix.
>
> We hope that this is what you envisioned with your question. We revised several parts of the paper during this change, and will post a description of all changes in the comment section. Again, thank you for your feedback.

---

### Official Review · AnonReviewer2 · 2020-10-28
**Interesting topic but the method seems to just be a feature extraction + fully connected layers.**

**Rating:** 3
**Confidence:** 3

**Review:**



This paper aims at defining a new architecture, Dynamic Recurrent Neural Network, that would be based on discrete differential equations of basic linear system transfer functions known from dynamic system identifiation. They show also an application example.

The paper is correctly written, and the subject is of interest. But I don't really see the point as it looks like the method removes all of what is 'recurrent', as no weights from the new unit are learned during the training. The learning is only on some fully connected layers afterwards: I feel that the authors just made an (interesting) 'feature extraction' instead of a recurrent unit, by using standard transfer functions without any weights to be learned (or haven't I understood correctly?).

The claim that the paper is the first to present a method where the sequence sampling rate, or 'delta t', is a parameter that can be modified without needed to re-train the network seems odd. In fact, I am not very familiar with this field but it looks to me that a lot of works that are now combining neural nets and dynamical systems are by definition able to do that. For example, I know this work 'Learning Dynamical Systems from Partial Observations', Ayed et al. 2019, which has a section called 'Benefits of Continuous-Time.', where I can read 'this allows us to accommodate irregularly acquired observations, and as demonstrated by the experiments, allows interpolation between observations.' So it is not something 'new' to the community and I would guess that papers closer to yours would also have the same feature?

Questions/ remarks:
- please explain better how your network is still recurrent : where is the 'hidden memory' that is passed? And where are the weights inside the DYRNN that are learned during the training? For me it looks like only the FC model placed after, that is moreover without any non-linearity (ReLU, etc), contains weights that are updated during training.
- 'and are state of the art layer types for text based sequence-to-sequence problems like machine translation or text processing --> no, the recurrent NN are not state of the art in translation anymore... transformers are. But RNN could be state of the art in other problems.
- 'The number of output channels per layer n layer amounts to (base component count * n oc = 5 * n oc ), as shown in Figure 3b.' --> not clear: there is 5 output channels, so noc = 5, but what is this 'base component count'? Why do we have to multiply it, and what is a layer? Or do you mean that there are 5 DYRNN units? or that every 'component' (P, I, ..) outputs the 5 outputs? This is not what is represented here.
- Fig. 6 might not be necessary, we can understand the concept without it.
- Modell --> Model

---

> ### Author Response · Authors · 2020-11-23
> **Response to AnonReviewer2**
>
> Thank you very much for reading our work and your thorough remarks. Your feedback showed us where further adjustments to the text were needed to do in order to improve comprehensibility for the reader and we have implemented adjustments of the text to try to reflect this.
>
> > The paper is correctly written, and the subject is of interest. But I don't really see the point as it looks like the method removes all of what is 'recurrent', as no weights from the new unit are learned during the training. The learning is only on some fully connected layers afterwards: I feel that the authors just made an (interesting) 'feature extraction' instead of a recurrent unit, by using standard transfer functions without any weights to be learned (or haven't I understood correctly?).
> > (...)please explain better how your network is still recurrent : where is the 'hidden memory' that is passed? And where are the weights inside the DYRNN that are learned during the training? For me it looks like only the FC model placed after, that is moreover without any non-linearity (ReLU, etc), contains weights that are updated during training.
>
> The network can be seen as a linear feature extractor. However the layer does contain weights, namely the K and T values from the linear equations. If you revisit equations (4) through (8) in the paper, you will see that some equations are based on their result from the prior timestep at (k-1). Therefore the hidden state of the network is [x(k-1), pt(k-1), k(k-1)] per cell.
>
> >The claim that the paper is the first to present a method where the sequence sampling rate, or 'delta t', is a parameter that can be modified without needed to re-train the network seems odd. In fact, I am not very familiar with this field but it looks to me that a lot of works that are now combining neural nets and dynamical systems are by definition able to do that. For example, I know this work 'Learning Dynamical Systems from Partial Observations', Ayed et al. 2019, which has a section called 'Benefits of Continuous-Time.', where I can read 'this allows us to accommodate irregularly acquired observations, and as demonstrated by the experiments, allows interpolation between observations.' So it is not something 'new' to the community and I would guess that papers closer to yours would also have the same feature?
>
> Thank you for providing this additional piece of related work. We are aware of methods using ODE solvers and have cited one of them in our Related Work section. Our work differs from these models because our model is trained in a sequence-to-sequence fashion on a complete measurement timeseries without the use of an ODE solver. Nonetheless, internally it works similarly to an ODE solver, since the equations are discretized using backwards Euler.
>
> >The number of output channels per layer n layer amounts to (base component count * n oc = 5 * n oc ), as shown in Figure 3b.' --> not clear: there is 5 output channels, so noc = 5, but what is this 'base component count'? Why do we have to multiply it, and what is a layer? Or do you mean that there are 5 DYRNN units? or that every 'component' (P, I, ..) outputs the 5 outputs? This is not what is represented here.
>
> We also adapted this part in the paper, seeing as it was written unclearly. What this section meant is that one DYRNN layer contains 5 basic subcomponents internally, by which we mean P, I, D, PT and PD.
> If the layer's input has one channel only, the layer output channel count is 5*1, i.e. 5 channels. If the layer's input is e.g. 3 channelled, the output results in 15 output channels in total.
>
> We have also added more changes to the paper, which we will describe in the general comment section.

---

### Author Response · Authors · 2020-11-23
**Response to all Reviewers**

Dear reviewers, thank you all for reading our paper and taking the time to write such helpful and constructive feedback. While extending the  interpretation for our model during this rebuttal phase, we detected a mistake in the implementation of one variant of the network, namely the one that is trained for one sampling rate at a time (i.e. the experiment from the current Figure 9 was unchanged). The mistake entailed that the hidden state of the I subcomponent was passed incorrectly. This showed us that there are actually two potential versions of our model: One with P, I, D, PD and PT which we call DYRNN5 and one with P, PD and PT called DYRNN3. Luckily, this mistake enabled us to see that often the model acutally improved if the amount of functions is reduced, i.e. if the model just contains P, PD, PT.
We want to apologize for the amount of changes that the reviewers have to reread and thank them for their patience while waiting for the rebuttal version. We hope that they will approve of our efforts to fix our mistakes and provide transparency on our new results.

Furthermore we implemented the following changes:
- Fixed writing errors and changed text to EN_GB
- Adjusted Chapter 3.1 to clarify the recurrent aspect of our model and what the weight parameters are
- Section 3.3 interpretation has been improved to explain how a differential equation can be derived from a trained model
- All prior experiments were redone
- The experiments were further explained and plots for all datasets were added
- New aspect of interpretability: By translating the trained model into the Laplace/Fourier domain, several properties can be visualized (e.g. its frequency response, root locus curve)

Thank you again for your help and for your patience while waiting for this new revision.

---

### Decision · Program_Chairs · 2021-01-07
**Final Decision**

**Decision:**

Reject

**Comment:**

This paper proposes a new RNN architecture called Dynamic RNN which is based on dynamic system identification.

Reviewers questioned the expressivity of the proposed model, practical application/impact of the proposed model, and interpretability of the proposed model. Even though the authors attempted to convince the reviewers, 3 out of 4 reviewers think that this work is not ready for publication.

Specifically, R4 recommends 5 ways to strengthen the paper. I recommend the authors to incorporate this feedback and make a stronger resubmission in the future.